# Identifying naturally occurring communities of primary care providers in the English National Health Service in London

Jonathan Clarke ,[1,2] Thomas Beaney ,[3] Azeem Majeed,[4] Ara Darzi,[5] Mauricio Barahona [2,6]

[1]Centre for Health Policy, Imperial College London, London, UK
[2]Centre for Mathematics of Precision Healthcare, Imperial College London, London, UK
[3]Department of Primary Care, Imperial College of Science Technology and Medicine, London, UK
[4]Primary Care, Imperial College, London, UK
[5]Imperial College London, London, UK
[6]Department of Mathematics, Imperial College London, London, UK

**Correspondence to**
Dr Jonathan Clarke;
j.clarke@imperial.ac.uk

## ABSTRACT

**Objectives** Primary Care Networks (PCNs) are a new organisational hierarchy with wide-ranging responsibilities introduced in the National Health Service (NHS) Long Term Plan. The vision is that PCNs should represent 'natural' communities of general practices (GP practices) collaborating at scale and covering a geography that fits well with practices, other healthcare providers and local communities. Our study aims to identify natural communities of GP practices based on patient registration patterns using Markov Multiscale Community Detection, an unsupervised network-based clustering technique to create catchments for these communities.

**Design** Retrospective observational study using Hospital Episode Statistics - patient-level administrative records of attendances to hospital.

**Setting** General practices in the 32 Clinical Commissioning Groups of Greater London

**Participants** All adult patients resident in and registered to a GP practice in Greater London that had one or more outpatient encounters at NHS hospitals between 1st April 2017 and 31st March 2018.

**Main outcome measures** The allocation of GP practices in Greater London to PCNs based on the registrations of patients resident in each Lower Layer Super Output Area (LSOA) of Greater London. The population size and coverage of each proposed PCN.

**Results** 3 428 322 unique patients attended 1334 GPs in 4835 LSOAs in Greater London. Our model grouped 1291 GPs (96.8%) and 4721 LSOAs (97.6%) into 165 mutually exclusive PCNs. Median PCN list size was 53 490, with a lower quartile of 38 079 patients and an upper quartile of 72 982 patients. A median of 70.1% of patients attended a GP within their allocated PCN, ranging from 44.6% to 91.4%.

**Conclusions** With PCNs expected to take a role in population health management and with community providers expected to reconfigure around them, it is vital to recognise how PCNs represent their communities. Our method may be used by policymakers to understand the populations and geography shared between networks.

### Strengths and limitations of this study

► In the absence of data-driven approaches, Primary Care Networks (PCNs) have formed through inter-personal relationships between practices rather than through an understanding of the distribution of their registered patients.

► This study uses Markov Multiscale Community Detection, a data-driven, unsupervised clustering method, to identify 'naturally occurring' communities of general practices (GP practices) to collectively form 165 PCNs across London.

► In doing so, this technique produces PCNs which are most representative of the spatial communities of patients for whom PCNs provide care.

► National Health Service England have proposed that PCNs should contain 30 000 to 50 000 patients restricted to a single Clinical Commissioning Group, however we find this may not represent patterns of care delivery in an urban setting.

► The use of Hospital Episode Statistics ensures that the obtained PCNs are related to secondary care utilisation; yet, on the other hand, these PCNs may not reflect patients who rarely use healthcare services but remain registered to a GP practice.

## INTRODUCTION

The introduction of Primary Care Networks in the National Health Service (NHS) Long Term Plan in 2019 marks one of the biggest changes to general practice in England.[1] Since the NHS was formed, general practice (GP practice) has seen significant evolution from predominantly single-handed GPs, to large-scale businesses incorporating multidisciplinary teams. This trend has resulted in a corresponding decline over time in the number of practices to 6984 practices in England in March 2019 with a corresponding increase in the average list size to 8550 patients.[2] In parallel, different organisational hierarchies have developed and coexist, linking together practices within a defined geography. Currently, many GP practices collaborate informally as Federations, while all GP practices are now part of larger Clinical

Commissioning Groups (CCGs). More recently, Sustainability and Transformation Partnerships have formed through collaboration between CCGs. These structures form part of a move towards integrated systems of care, with the aim of reducing the complexity and fragmentation within healthcare delivery. There is a lack of robust evidence on the effects on patient outcomes, but some evidence indicates that integrated care may improve patient satisfaction and may reduce hospital admission rates and length of stay in older adults.[3 4]

The new NHS Long Term Plan and GP contract announced in 2019 introduced the 'Primary Care Network' (PCN),[1 5 6] a major new organisational structure operating at a population size smaller than the CCG. PCNs are designed to provide economies of scale, without losing the community focus of general practice. In contrast to the informal arrangement of GP Federations, PCNs represents a policy shift towards more formal collaboration between GP practices at a smaller scale than CCGs. This new organisational structure comes with significant financial incentives—an additional £1.8 billion in the form of Directed Enhanced Services has been earmarked to recruit additional staff and expand the role of primary care.[7] Two of the key roles of PCNs will be to integrate community care providers within the network, and to take on an increasing role in the population health management of their communities. These objectives highlight the importance of properly delineating geographically contiguous areas to ensure the efficiency of services provided by PCNs.[5]

While many practices are already members of networks or federations, or informally collaborate with other GPs, the new organisational structure may not match existing arrangements. According to NHS England, PCNs are specified to be networks of neighbouring practices covering a population of 30 000 to 50 000, which will not cross the boundaries of a CCG, although this is not an absolute requirement.[5] The vision from NHS England is that PCNs serve 'natural communities' and that their boundaries make sense to their practices, community-based providers and to their local communities.[8] In reality, boundaries that make sense to all three groups may be difficult to harmonise, particularly where patients are free to choose a GP outside their geographical area.

In this article, we set out an approach to defining communities that conform to the criteria required of PCNs, based only on the registration of patients from a given geographical area to a GP. Our approach uses Markov Multiscale Community Detection (MMCD), which uses Louvain optimisation to detect and obtain robust partitions of a network, without imposing a priori the number of partitions that should be produced.[9–11] Briefly, MMCD is an unsupervised graph-based clustering method that exploits how a diffusion process spreads on a network to identify communities of nodes that share information consistently over different time scales. In this case, the nodes are GP practices and the communities correspond to practices that have stronger connections with one another than to other nodes in the network. These techniques have previously been applied to the organisation of planned orthopaedic care in England[12] and, in other disciplines, to identify communities within processes as diverse as metabolic networks, transport systems and power grids.[12–16] In this study, we use these unsupervised network analysis techniques to identify underlying communities of GP practices who share patients from the same LSOAs of residence as one another. The ensuing 'GP communities' are thus identified in an unsupervised manner based on patient registration data. In doing so, we seek to produce and understand data-driven PCNs within London that meet the ambitions of policymakers to create 'natural' communities of primary care that represent their local populations.

## METHODS

All adult patients presenting to outpatient secondary care in England from 1st April 2017 to 31st March 2018 were identified from Hospital Episode Statistics (HES).[17] The Lower Layer Super Output Area (LSOA) of residence of the patient and the unique identifier of their registered GP practice were identified from these records. LSOAs are mutually exclusive, collectively exhaustive geographical fine-scale census divisions defined by the UK Office for National Statistics. There are 4835 LSOAs in London with a mean population of 1842 people. LSOAs are therefore similar in scale to Census Block Groups in the USA. The LSOA is the most granular geographical division which can be readily mapped to HES.

In cases where a patient was registered at more than one LSOA and GP practice combination within the 1 year time period of our study, the record with the highest frequency for that individual was chosen; where these were tied, the most recent combination was chosen. GP practices contributing fewer than 100 patients were excluded.

In order to quantify the extent of overlap between areas covered by different GP practices in London, the equivalent market size (EMS) of each LSOA was calculated as the reciprocal of the Herfindahl-Hirschman Index (Equations 1 and 2):

$$HHI_i = \sum_{j=1}^{N} s_{ij}^2 \qquad (1)$$

**Equation 1:** Herfindahl-Hirschman Index of LSOA $i$. Here $s_{ij}$ is the proportion of patients from LSOA $i$ registered to GP practice $j$, and $N$ is the number of GP practices in the data set.

$$EMS_i = 1/HHI_i \qquad (2)$$

**Equation 2:** Equivalent Market Size of LSOA $i$ is calculated as the reciprocal of the Herfindahl-Hirschman Index of LSOA $i$.

The EMS represents the number of GP practices to which members of an LSOA would be registered if each practice occupied an equal market share of registrations.

It therefore provides an estimate of the concentration of the market for primary care registrations within each LSOA.[18–20] The probability of a patient from each LSOA being registered to a GP practice was calculated, and the cosine similarity matrix between all pairs of GP practices was computed (Equation 3).

$$similarity_{AB} = \frac{\sum\limits_{i=1}^{n} A_i B_i}{\sqrt{\sum\limits_{i=1}^{n} A_i^2} \sqrt{\sum\limits_{i=1}^{n} B_i^2}} \qquad (3)$$

**Equation 3:** Calculation of the cosine similarity between two GP practices, A and B. Here $A_i$ is the proportion of patients registered to GP practice A resident in LSOA $i$; $B_i$ is the proportion of patients registered to GP practice B resident in LSOA $i$; and $n$ is the total number of LSOAs in the data set.

Equation 3 defines one of the entries of a similarity matrix between all GP practices. This similarity matrix can be thought of as the adjacency matrix of a network connecting GPs to one another with weights reflecting the similarity of their patterns of patient registration across LSOAs, that is, GP practices with similar LSOA patient profiles are strongly connected. This dense network was sparsened using the Relaxed Minimum Spanning Tree (RMST) technique, a method used elsewhere in applied network science to sparsen a dense, inhomogeneous network to preserve both local and global connectivity within a network.[21–23] The sparsened network was subsequently partitioned using MMCD to produce partitions of the GPs according to shared patterns of registration from LSOAs. Each of these partitions corresponds to a 'natural' community directly derived from the registration data.[9 10 12 24]

MMCD was performed over a range of 400 Markov times from 0.01 to 1 and over a range of 40 RMST pruning parameters from 2.1 to 6.0. Each of these 16 000 models was optimised 500 times and the most common network partition for each model was selected. Scanning across the range of pruning parameters revealed relative maxima with respect to our coverage function (described below), with clear surrounding margins of suboptimal values.

The geographical co-ordinates of each GP practice were identified from their registered postcode. For each of the 16 000 partitions produced, the pairwise geographical straight-line distances between all practices within each PCN was calculated. Where the median distance from a practice to all other practices within a PCN community was more than four times the median pairwise distance between all practices within the PCN community, this practice was excluded as a spatial outlier. The number of practices within each PCN community was calculated and the proportion of practices contained within the polygon drawn around the outer geographical limits of the practices comprising the PCN community was calculated.

PCN communities where the number of practices was less than 3 or greater than 20, or where the percentage of

practices spatially located within the outer spatial limits (defined by the polygon drawn around each practice coordinate) of the PCN community was more than 25% were excluded. The total number of practices present in the remaining PCN communities was calculated. The partition with the highest number of included GP practices was taken as the optimal partition.

For this optimal partition, the GP practice name, location and CCG were linked by practice code, using data from NHS Digital. Practice list sizes as of March 2018 were also linked using data from NHS Digital.[2] Where a practice list size was not available, such as in cases where a practice had closed or merged over the time period, list sizes as of April 2017 were used.[25]

LSOAs were subsequently assigned to a PCN community based on the PCN to which the highest number of patients within a given LSOA were registered. GP practices were mapped along with their corresponding assigned community allocation and LSOA boundaries. The proportion of patients resident within the same PCN as their registered GP was calculated for each PCN.

### Patient and public involvement statement

The data on which this study is based was granted following review by a panel including patient and lay representatives. Patients were not invited to contribute to the writing or editing of this document for readability or accuracy.

## RESULTS

A total of 3 428 322 unique adult patients resident in London attended a total of 20 173 937 outpatient appointments to NHS hospitals between 1st April 2017 and 31st March 2018 (figure 1). All 4835 LSOAs in London were represented. A total of 1334 GP practices were identified within London belonging to a total of 32 Clinical Commissioning Groups.

Residents in the same LSOA in London were generally registered to a wide range of GP practices. Figure 2 demonstrates the equivalent market size of LSOAs across London with respect to GP practices. A wide range of equivalent market sizes ($EMS_i$) was seen, from 1.1 to 20.6. 185 LSOAs (3.8%) had an equivalent market size of less than two GP practices, consistent with primary care provision by a single dominant GP practice. The median equivalent market size across London was 4.9 GP practices per LSOA, while 259 LSOAs (5.4%) had an equivalent market size of more than 10 GP practices. The median LSOA-level equivalent market sizes of CCGs in London ranged from 3.2 to 7.0. Overall, the median equivalent market size for LSOAs north of the river Thames was 23% higher than those south of the river (5.3 vs 4.3).

An optimal configuration of GP communities was found at an RMST parameter of 5.5 and Markov time of 0.054. For this optimal clustering, only 43 GP practices (out of a total of 1334) were unassigned to a community according to our criteria: 28 practices lay in 20 communities with

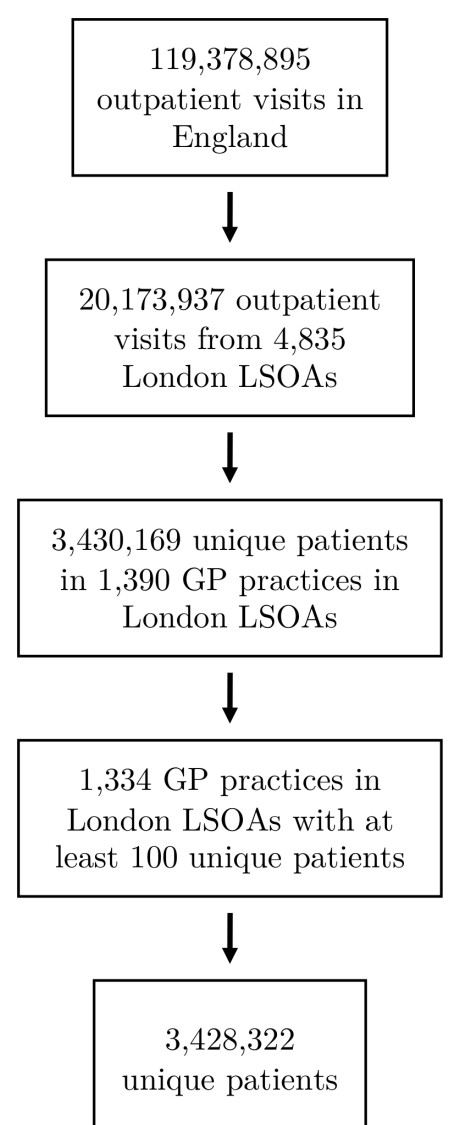

**Figure 1** Algorithm for participant selection from Hospital Episode Statistics. GP practices, general practices; LSOA, Lower Layer Super Output Area.

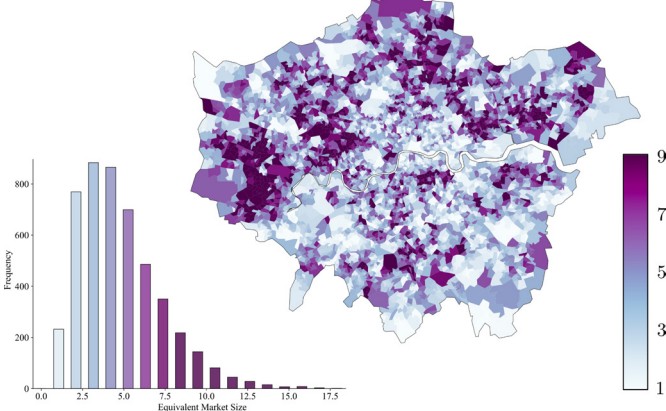

**Figure 2** Choropleth map representing the equivalent market size of general practices for all LSOAs in London, as calculated in Equation 2. The histogram shows the frequency distribution of equivalent market sizes for the 4835 LSOAs in London. Higher values indicate more dilute markets with patients registered to a wider range of GP practices. GP practices, general practices; LSOA, Lower Layer Super Output Area.

from only one CCG and the remaining PCNs contained practices from either two (23.0%) or three (9.1%) CCGs.

Across all 4721 LSOAs, the median percentage of patients registered to GP practices located within their allocated PCN was 73.7%, ranging from a minimum of 24.8% to a maximum 98.6% (figure 4, left). Across the 165 PCNs, a median of 70.1% of patients within the PCN catchment area were registered to a GP practice within the PCN, ranging from a minimum of 44.6% to a maximum

only 1 to 2 practices, and a further 15 practices were spatial outliers. Collectively, these 43 practices were the modal provider of primary care for 114 LSOAs with a total population of 187 101 (2.3%). Our optimal partition consisted of 165 PCNs grouping 1291 practices, which cover 4721 LSOAs in London and 97.7% of the estimated London population. A map of this optimal configuration, displaying GP practices superimposed on the LSOAs assigned to each PCN is shown in figure 3. The excluded communities, represented by dashed lines, are predominantly situated peripherally or bordering the river Thames.

The PCNs ranged in size from 3 to 18 practices with a median of 8. Median list size of PCNs was 53 490 patients (ranging from 14 574 to 176 982 patients, with a lower quartile of 38 079 and upper quartile of 72 982 patients). Around two-thirds (67.9%) of PCNs contained practices

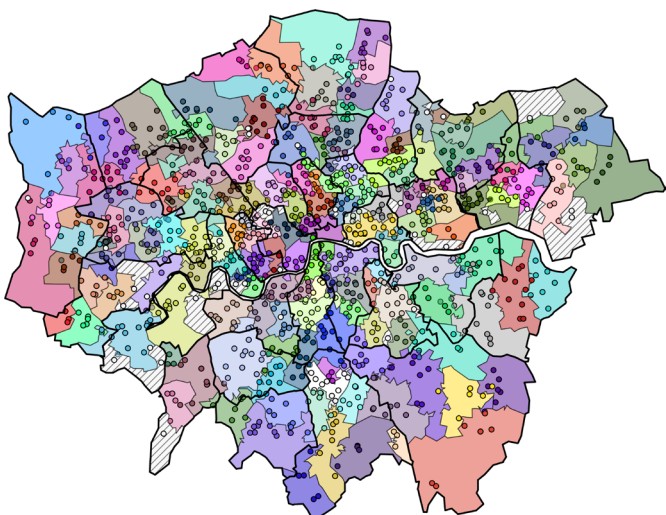

**Figure 3** Optimal configuration of general practice communities in London obtained from our model, with LSOAs assigned to most common practice of registration. Dashed lines represent unassigned communities. Circles represent the location of GP practices, and are coloured according to the community they are assigned. White circles indicate the 43 GP practices that are not assigned to any community. GP practices, general practices; LSOA, Lower Layer Super Output Area.

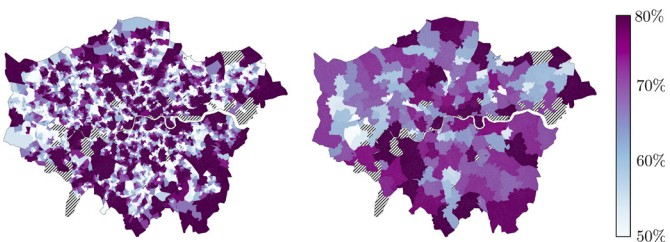

**Figure 4** Left: Choropleth map representing the proportion of patients within each LSOA attending the modal PCN. Right: Choropleth map representing the proportion of patients within the LSOAs assigned to each PCN attending a practice within the PCN. Dashed lines represented unassigned communities. LSOA, Lower Layer Super Output Area; PCN, Primary Care Network.

of 91.4% (figure 4, right), and this did not differ between PCNs that did or did not cross CCG boundaries (70.0% vs 70.2%, two-tailed Mann-Whitney U test, p=0.36). The median Herfindahl-Hirschman Index of the LSOAs was 0.57, and there was a median equivalent market size of 1.75 PCNs for each LSOA.

## DISCUSSION

As the health system develops to prioritise integrated and collaborative working, we need to improve our understanding of the relationships between providers and patients. With the requirement for GPs to form larger organisational units in the form of PCNs, there is a need to quantify how representative these communities are of their populations. The techniques used in our study provide an unsupervised, data-driven means of producing mutually exclusive PCNs formed by bringing together GP practices that frequently provide care to patients from the same geographical regions. In doing so, it makes no assumptions regarding geography or organisational hierarchies and produces 'natural' communities. Using this technique, we showed that despite having no prior geographical knowledge, 97.7% of the population of London may be assigned to a PCN of appropriate size and which are spatially consistent.

### The composition of PCNs

The factors that determine practices joining a PCN are complex and rely on the interpersonal and professional relationships between GPs as much as the shared relationships in registration of the local population. Furthermore, predefined boundaries cannot be ignored, with local authorities and CCGs taking responsibility for commissioning for a given geographical area and registered patients. However, networks must work for their population, and existing tools to guide decision-making are limited. While most practices have now joined PCNs, it is likely that boundaries will change as practices move in and out of PCNs, reflecting the complex dynamics of a system which needs to work for the population and providers.

In the context of the PCNs that are forming across London, our findings may in the future be directly compared with the actual PCNs that form. While it is hoped that the PCNs suggested by this study conform to those being formed across London, identifying discrepancies between the two conformations may offer a means of identifying PCNs which may not adequately represent local patterns of patient registration, and therefore may benefit from reorganisation. Second, where practices struggle to align with one another to form PCNs, the findings of this study offer a tool by which to propose suitable PCN structures.

Our findings also raise questions regarding the optimal size and configuration of PCNs across London. Our modelling suggests that the 30 000 to 50 000 recommended list size for PCNs may be too restrictive. The finding of a median list size for our optimal configuration of 53 490 in London, with an IQR from 38 000 to 73 000 patients suggests that the current recommended range may not permit larger networks to form where underlying patient registration patterns would favour this. A wider range of list sizes also suggests that greater variability in PCN size should be allowed depending on local need. Similarly, while a clear majority of the PCNs we found (67.9%) are formed from practices within a single CCG, almost one-third contain practices in two or three CCGs, suggesting that there should be flexibility in conformity to CCG boundaries. However, there may be additional problems for a PCN containing practices from more than one CCG where priorities do not align.

### The role of PCNs

If PCNs take on a role in population health management, and community services are required to reconfigure to match the footprints of PCNs,[26 27] then the need to understand the demographics and geography of PCNs becomes critical. Through our network analysis method, we defined an approach to assign catchment areas and estimate the catchment population for each PCN. We found that on average, almost one-third of the population living within the PCN catchment area attended a practice outside of the PCN, which was as high as half of the population for some PCNs. Understanding the dynamics of registration patterns in the local population could thus have significant implications on the design of place-based services.

Population dispersion is likely to be a greater problem in a densely populated urban area such as London, where there are extensive transport links and a greater choice of providers within a shorter commuting distance. However, our estimates, which by design are modelled to represent optimal patterns of registration, are likely to overestimate coverage of the PCN compared with those PCNs that have already formed. The opportunity for patients to retain registration with a practice after moving outside of its historic catchment postcodes may reduce coverage of a PCN where a fixed relationship to a discrete geographical area is required.

The emergence of digital NHS general practice, in the form of GP at Hand and its peers which allow patients access to video consultations as the first point of contact, disrupts the notion of a 'place-based' relationship between primary care providers and geographical communities of patients. Currently, the registered population for such services is disproportionately of working age, with 98.5% of registrants between 20 to 64 years of age, compared with a London-wide estimate of 75.3%.[28 29] The proportion of patients with one or more long-term conditions using this service is almost half that of London-wide estimates (10.6% compared with 18.3%).[28 30] Therefore, patients registered with digital general practices are less likely to require the place-based services and community integration which underpins the establishment of PCNs.

As such innovations in registration and GP provision mature and potentially scale across the health system, the significant differences between patients using digital primary care services and traditional general practices may undermine efforts to enforce relationships between primary care providers and discrete geographical communities of patients. This trend may signal the emergence of a differentiated system of primary care, where those with low care needs are served by essentially rootless digital primary care providers and those with higher care needs are attended to by well-integrated, accessible primary care providers with a nearby physical presence. In such cases, the creation of PCNs by providers of digital primary care may be orthogonal to the underlying ethos of vertical integration and investment in relationships with community care providers that underlies the current policy.

### Limitations

One of the main limitations of our analysis is in use of secondary care data, rather than primary care data. While HES represented 3.4 million patients in London, this covers less than half of the estimated population of 8.8 million.[27 31] Patients in HES have, by definition, used secondary care services at least once over the study period, and may not be representative of the whole population registered with primary care. Older people, and those with a higher number of chronic conditions are more likely to be represented in HES, and these patients may have different criteria in choosing a GP. Use of complete lists of GP practice registrations obtained from practices and covering the whole of London, would be preferable, but is not readily available and was outside the scope of the current study.

A limitation with our method was that the optimal configuration was unable to fit every practice in London to a PCN, with 43 (out of 1334) unassigned as a result of our selection criteria. Fifteen practices were unassigned as spatial outliers, which is to say that their median distance to all other practices in their network was more than four times greater than the median pairwise distance between all practices in the network. These rare instances may reflect the statistical noise of the modelling technique which is agnostic to the spatial proximity of providers

to one another. A further 28 practices were unassigned due to their proposed PCNs containing fewer than three practices. In these cases, allocation of unassigned practices in collaboration with practices and commissioners, to nearby larger PCNs could be an appropriate solution to ensure complete allocation of practices to PCNs. The finding that many unassigned practices were near the periphery of London suggests a boundary effect where the exclusion of practices and the population outside of London may have affected the model in these regions.

### CONCLUSION

As health systems adapt towards closer integration across services, network analysis offers a data-driven and unbiased means of understanding the connections between PCNs and their patients. Our findings demonstrate that GP practices may be combined into communities reflecting their underlying populations in accordance with the specification of PCNs. At a time when integration of community, primary and secondary care is being prioritised, concurrently, place-based primary care anchored in the local community is increasingly being challenged with the growth of online GP consultation providers, such as that provided by GP at Hand in London. Upscaling primary care into larger networks has the potential to weaken further the ties between providers and their communities. There is a pressing need to better understand how these networks will represent their geographies and patients, to identify who may gain and who may lose out, and ensure a well-intentioned policy does not widen inequalities in health.

**Acknowledgements** AM is a GP Principal in an NHS general practice.

**Contributors** JC and TB were involved in all aspects of the study. MB was involved in the development of the methodology and assisted in the formal analysis. MB, AM and AD were involved in the conceptualisation of the study and in the reviewing and editing of the draft. JC has had access to all the data in the study and all authors had final responsibility for the decision to submit for publication. JC attests that all listed authors meet authorship criteria and that no others meeting the criteria have been omitted.

**Funding** This article is independent research supported by grants from The Peter Sowerby Foundation and the National Institute for Health Research (NIHR) Imperial Patient Safety and Translational Research Centre (PSTRC) PSTRC_2016_004. Infrastructure support for this work was provided by the NIHR Imperial Biomedical Research Centre (BRC) 1215-20013. TB is supported by a National Institute for Health Research (NIHR) Academic Clinical Fellowship. TB acknowledges support from the NIHR Imperial Biomedical Research Centre (BRC) and the NIHR Applied Research Collaboration for NW London. AM is supported by the NIHR Applied Research Collaboration for NW London. MB acknowledges support from EPSRC grant EP/N014529/1 supporting the EPSRC Centre for Mathematics of Precision Healthcare. Data management was provided by the Big Data and Analytical Unit (BDAU) at the Institute of Global Health Innovation (IGHI), Imperial College London. The views expressed in this publication are those of the author(s) and not necessarily those of the NHS, the National Institute for Health Research or the Department of Health.

**Map disclaimer** The depiction of boundaries on this map does not imply the expression of any opinion whatsoever on the part of BMJ (or any member of its group) concerning the legal status of any country, territory, jurisdiction or area or

of its authorities. This map is provided without any warranty of any kind, either express or implied.

**Competing interests** All authors have completed the ICMJE uniform disclosure form at www.icmje.org/coi_disclosure.pdf and declare: JC, TB, AD and MB have no competing interests to declare.

**Patient and public involvement** Patients and/or the public were not involved in the design, or conduct, or reporting, or dissemination plans of this research.

**Patient consent for publication** Not required.

**Ethics approval** This study received local ethical approval through the Imperial College Research Ethics Committee (17IC4178).

**Provenance and peer review** Not commissioned; externally peer reviewed.

**Data availability statement** Data are available upon reasonable request. Data used in this study were obtained from NHS Digital for the purpose of this work and may only be accessed through direct application to NHS Digital. Patient-level data is required for the analyses conducted, and therefore sharing of data pertaining to this study is not possible. Data for primary care network assignments are available from the authors on request. Data as to GP practice-level allocation to putative Primary Care Networks will be made publicly available on www.healthdatascience.co.uk and have been disseminated to local primary care providers and commissioners.

**ORCID iDs**
Jonathan Clarke http://orcid.org/0000-0003-1495-7746
Thomas Beaney http://orcid.org/0000-0001-9709-7264
Mauricio Barahona http://orcid.org/0000-0002-1089-5675

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
