## [Reviewer comments · BMJ Open]

ARTICLE DETAILS

TITLE (PROVISIONAL)	Identifying Naturally Occurring Communities of Primary Care Providers in the English National Health Service in London
AUTHORS	Clarke, Jonathan; Beaney, Thomas; Majeed, Azeem; Darzi, Ara; Barahona, Mauricio

VERSION 1 - REVIEW

REVIEWER	Christoph Strumann University of Luebeck, Germany
REVIEW RETURNED	12-Feb-2020

GENERAL COMMENTS	Report on "Identifying Naturally Occurring Communities of Primary Care Providers in the English National Health Service in London" Manuscript Number: bmjopen-2019-036504, BMJ Open The study proposes an approach to identify underlying 'natural' communities of GP practices. The authors apply unsupervised, data-driven network analysis techniques to produce mutually exclusive networks of GP practices that frequently provide care to patients from the same spatial area. The results of the study are well described and discussed. In what follows, I give a couple of suggestions that may improve the paper. Introduction • The introduction briefly describes the legislative need for the GPs to organize Primary Care Networks (PCN). However, could you also motivate the need for a closer integration by referring to the relevant literature showing the gains for the population health if primary care physicians are closer integrated? Methods • The approach you applied is somewhat hard to follow according to your descriptions. It might be very helpful if you briefly describe the main idea of the methodological approach at the beginning of the Methods section. In the current version of your manuscript, it is e.g. not clear, why you need to compute the equivalent market size of each LSOA? Is it used for the similarity (what I do not think so) or the determination of the size of the PCN? Or is it just used to describe the current distribution of the primary health care provider?
--

	 • Instead (or additionally) to the HHI it will be helpful for the reader if you provide the explicit formula how the equivalent market size of each LSOA is obtained/estimated. • If the HHI is computed for each LSOA, you might use indices to indicate the LSOA-level. • For the international reader it might be helpful if you briefly state why you use LSOA as the primary spatial unit. • In line 11, the index l might be substitute by i, i.e. LSOA_{i}. Results  • The figures are not easy to understand. It would be very helpful if you provide additionally to the captions also notes or legends. For instance, the circles in Figure 2 might be the GP practices. However, what are the white circles? Moreover, how did you mark the networks? Discussion  • The discussion is very well done. Especially addressing the increasing demand for e-health services. However, these aspects might also reveal that homogeneity among the physicians patient pool might be a relevant measure for determining optimal physician networks. Would it be possible to weight the patients accordingly to their characteristics, e.g. age, health needs, multimorbidity? This might result in PCNs with more homogenous shared patient pools? • In this sense, you might weaken the limitation that your approach is based only on patients that used secondary care services.
--	--

REVIEWER	Erika Moen Geisel School of Medicine at Dartmouth United States
REVIEW RETURNED	24-Feb-2020

GENERAL COMMENTS	This manuscript identified naturally occurring networks of primary care practices based on patient registration patterns using Markov Multiscale Community Detection, an unsupervised network-based clustering technique. By creating catchments for these primary care network communities, the results may help shape British national policy regarding a new organizational hierarchy of primary care networks (PCNs). The method used is likely to be of broad interest to health services and policy researchers, particularly as health systems are being reorganized to optimize both formal and informal relationships to improve quality of care. The manuscript is well-written. The critiques and suggestions I have are enumerated below.  1. Page 4 lines 26-30: the purpose/function of LSOAs might be briefly described here 2. Page 6 line 32: it would be helpful for the authors to justify pruning parameters 3. Page 6 lines 37-40: "pairwise distances" ...it was not immediately
--

apparent that this referred to geographic distances (as opposed to cosine similarity, for example). Is physical address being used?

4. Page 6 lines 48-52: "with the outer spatial limits" - define?

5. Page 7 lines 44-59 are not accessible to readers who are unfamiliar with the neighborhoods of London. Can the Figures (maps) be labelled in a way that corresponds to the text to orient a broader audience to the results being presented in that paragraph?

6. Page 7 lines 55-59: how can we interpret the within LSOA effect of proportion of 65+ patients on EMS without taking into account the population size? Small populations are likely to have a HHI closer to 1, which would correspond to a low EMS.

7. Page 8 lines 24-36: were PCNs with lower percentage of patients registered to GP practices located within the PCN more likely to span more than one CCG?

8. Page 9 lines 14-24: while the location of patient LSOA relative to practice location was considered, the attribution of patients to a single practice removes the information that examining patient-sharing between practices might provide.

9. Page 9 lines 27-40: is the PCN list size not also a product of the maximum number of allowed practices in a PCN (20)...and the fact that the partition was optimized for the highest number of practices?

10. Page 10 lines 13-24: GP at Hand should be described, briefly.

11. Page 10 lines 57-58: "Use of GP registration data would be preferable, but is not readily available..." The meaning of GP registration data in this sentence is unclear. Didn't this study use GP registration data?

12. Page 10 lines 55-57: I'm not sure that the EMS numbers can reflect patient choices without considering practice size and LSOA population

13. Page 11: conclusion seems less derived from data and more a commentary on seemingly unaddressed aspects of the online providers, the current PCNs, and the size of CCGs. Can this be written in a way that connects the commentary to the results more directly? Or provide a clearer rationale for why these topics should be addressed in the conclusion?

14. Page 14: figure legends should be more descriptive. The figures are not informative to readers without more context.

15. Fig 2: what are the color scales supposed to represent? The scales appear to have different upper bounds in the two images, but the figure implies the scale represents EMS.

Minor comments

Page 2 lines 32 and 36: the acronym LSOA is never introduced

Page 4 line 5: should the acronym "NHS" be introduced here formally?

Page 4 line 8: should the acronym "GP" be introduced formally here?

Page 6 line 10: "presentations" is unclear.

	Possible typo in the legend of Figure 2: "LSOA I" should probably read "LSOA i"
--	---

REVIEWER	Marieke Perry Radboudumc, Nijmegen, The Netherlands
REVIEW RETURNED	10-Mar-2020

GENERAL COMMENTS	This study aims to identify natural communities of GPs based on patient registration patterns as a reaction on the NHS Long Term Plan of Primary Care Network formation. I consider the study design using big data an interesting and original approach to achieving this aim. Since the quality of interprofessional relations is essential in network collaborations, I wonder whether quantitative study alone can service PCN formation. My statistical knowledge is insufficient to judge the appropriateness of the analyses conducted. I therefore advise an additional statistical review for the study. Please find below some more detailed comments: In general, abbreviations are not always appropriately introduced and used throughout the paper. Abstract: As a reviewer with limited knowledge of the British healthcare system, the abstract was initially difficult to understand. After reading the full article, everything fell more into place. The abstract would gain clarity when a little more information could be provided on the PCNs. Furthermore, a short explanation of what relevant data these hospital statistics contain. It is mentioned between the lines in the 'main outcome measures section' but it could be made more explicit. Introduction: Well written, clear explanation of NHS policy. Mainly written from a policy perspective. As a clinician, I wonder whether GP could give you the information on the catchment areas of their interprofessional collaboration if you would simply ask them. Methods: Original big data approach, clearly written, but my statistical knowledge is insufficient to judge the appropriateness of the analyses. You describe that outliers are excluded with regard to distance to other practices in a PCN and with regard to PCN size. From a statistical point of view, I understand the choice for exclusion. From a policy point of view, exclusion of practices does not seem a possibility. The outliers in my view do request additional (qualitative) research, aimed at understanding the mechanisms to the development of these clusters. These considerations could be included in the discussion section. The fact that the study is performed in hospital registrations is in my opinion a major limitation which is considered in the discussion. If the readership that you want to reach with this paper are policy makers and may be also primary care professionals I consider the Equation shown of no additional value. I would consider including these in an Appendix. Results:
---

	The median percentages presented include broad ranges which I would like to see considered in the discussion section. Discussion: In the summary of the results, you mention that '97.7% of the population can be assigned to a PCN of appropriate size and which are spatially consistent'. To my understanding, this is after the outliers were excluded. How would these results change when these excluded practices were allocated to PCNs as would happen in real life? The limitation raised on the data source (hospital registrations) I consider a serious limitation. The fact that these data represent more vulnerable patients, because they visited the hospital at least once during the last year, importantly influence the outcomes and would probably only increase the ranges to the median percentages reported. The described mechanism through which the outcomes could be influenced, could be linked to the paragraph on digital NHS general practices. I would exclude the remark that use of GP data 'was outside the scope of the current study' as you consider the use of GP data preferable.
--	--

VERSION 1 – AUTHOR RESPONSE

Reviewer 1:

Introduction

- The introduction briefly describes the legislative need for the GPs to organize Primary Care Networks (PCN). However, could you also motivate the need for a closer integration by referring to the relevant literature showing the gains for the population health if primary care physicians are closer integrated?

Thank you, and we agree that some rationale here is useful. We have added two sentences to the opening paragraph to summarise the rationale (for which hard outcomes are lacking), including two references.

Methods

- The approach you applied is somewhat hard to follow according to your descriptions. It might be very helpful if you briefly describe the main idea of the methodological approach at the beginning of the Methods section. In the current version of your manuscript, it is e.g. not clear, why you need to compute the equivalent market size of each LSOA? Is it used for the similarity (what I do not think so) or the determination of the size of the PCN? Or is it just used to describe the current distribution of the primary health care provider?

It is used to describe the extent of overlap between the areas covered by different GP practices. – changed in the text, methods paragraph 3

- Instead (or additionally) to the HHI it will be helpful for the reader if you provide the explicit formula how the equivalent market size of each LSOA is obtained/estimated.

This equation is now included beneath the HHI equation

- If the HHI is computed for each LSOA, you might use indices to indicate the LSOA-level.

Thank you. Equations 1 and 2 are now changed to include LSOA-level indices for clarity.

- For the international reader it might be helpful if you briefly state why you use LSOA as the primary spatial unit.

Thank you. We have included a couple of sentences in the first paragraph of methods to reflect this and provide international context.

- In line 11, the index I might be substitute by i , i.e. LSOA _{i} .

Thank you. We have made this change.

Results

- The figures are not easy to understand. It would be very helpful if you provide additionally to the captions also notes or legends. For instance, the circles in Figure 2 might be the GP practices. However, what are the white circles? Moreover, how did you mark the networks?

Thank you, we have made changes to the captions to clarify these. We hope that this addresses each of these questions. The caption for Figure 3 also explains that the LSOAs are assigned to the most commonly registered practice.

Discussion

- The discussion is very well done. Especially addressing the increasing demand for e-health services. However, these aspects might also reveal that homogeneity among the physicians patient pool might be a relevant measure for determining optimal physician networks. Would it be possible to weight the patients accordingly to their characteristics, e.g. age, health needs, multimorbidity? This might result in PCNs with more homogenous shared patient pools? In this sense, you might weaken the limitation that your approach is based only on patients that used secondary care services.

Thank you for this suggestion and this would be possible to some extent, as we know the age and gender profile within each LSOA, and could incorporate broad estimates of co-morbidities at larger geographic granularity. However, crucially, we would not know to which practice these individuals attended, which would require us to extrapolate from those attending secondary care at least once to those not attending at all. Given these may be very different population subgroups in terms of desired proximity to GPs, any analysis would still be subject to the same issue and likely not be worth the additional complexity.

Reviewer: 2

Reviewer Name: Erika Moen

Institution and Country: Geisel School of Medicine at Dartmouth

United States

Please state any competing interests or state 'None declared': None declared

Please leave your comments for the authors below

This manuscript identified naturally occurring networks of primary care practices based on patient registration patterns using Markov Multiscale Community Detection, an unsupervised network-based clustering technique. By creating catchments for these primary care network communities, the results may help shape British national policy regarding a new organizational hierarchy of primary care networks (PCNs). The method used is likely to be of broad interest to health services and policy

researchers, particularly as health systems are being reorganized to optimize both formal and informal relationships to improve quality of care. The manuscript is well-written. The critiques and suggestions I have are enumerated below.

1. Page 4 lines 26-30: the purpose/function of LSOAs might be briefly described here

Thank you – changes made in the text to reflect this.

2. Page 6 line 32: it would be helpful for the authors to justify pruning parameters

Thank you – justification is provided in the methods.

3. Page 6 lines 37-40: “pairwise distances” ...it was not immediately apparent that this referred to geographic distances (as opposed to cosine similarity, for example). Is physical address being used?

Thank you – changes made in the text to clarify this

4. Page 6 lines 48-52: “with the outer spatial limits” - define?

Thank you – we have added ‘located within the outer spatial limits (defined by a polygon drawn around each coordinate) of the PCN’ to clarify this.

5. Page 7 lines 44-59 are not accessible to readers who are unfamiliar with the neighborhoods of London. Can the Figures (maps) be labelled in a way that corresponds to the text to orient a broader audience to the results being presented in that paragraph?

Thank you. In light of your comments, we felt that focussing on specific CCGs may not be relevant to a wider audience and have therefore amended to text to focus on more general trends across London which are less reliant on specific geographic knowledge of the city.

6. Page 7 lines 55-59: how can we interpret the within LSOA effect of proportion of 65+ patients on EMS without taking into account the population size? Small populations are likely to have a HHI closer to 1, which would correspond to a low EMS.

Reflecting on this important comment, we have decided that the relationship between age and market size forms part of a wider analysis of the sociodemographic characteristics of regions and their healthcare markets. This does not have sufficient room to be expressed here, and we have therefore removed it to cover more it widely in a separate manuscript.

7. Page 8 lines 24-36: were PCNs with lower percentage of patients registered to GP practices located within the PCN more likely to span more than one CCG?

Thank you. We have included reference to this analysis in the results section. We found a small difference between median percentages for PCNs that do not cross a CCG boundary (70.2%) and do cross a boundary (70.0%), however this was not significant in a two-tailed Mann-Whitney U-test ($p = 0.36$).

8. Page 9 lines 14-24: while the location of patient LSOA relative to practice location was considered, the attribution of patients to a single practice removes the information that examining patient-sharing between practices might provide.

Patients are only registered to a single practice at a time, therefore patient-sharing between primary care practices is rare.

9. Page 9 lines 27-40: is the PCN list size not also a product of the maximum number of allowed practices in a PCN (20)...and the fact that the partition was optimized for the highest number of practices?

The partitions were optimised to permit a range of practice sizes in line with proposed PCNs. Configurations with smaller partitions did not produce optimal configurations due to the extent of overlap between practices. No PCNs contained the maximum number of practices, and the median PCN size was 8, a value well below the potential maximum. Changes made in the text to clarify this.

10. Page 10 lines 13-24: GP at Hand should be described, briefly.

A brief comment is inserted to describe this.

11. Page 10 lines 57-58: "Use of GP registration data would be preferable, but is not readily available..." The meaning of GP registration data in this sentence is unclear. Didn't this study use GP registration data?

We refer here to dedicated primary care level data, and have clarified this

12. Page 10 lines 55-57: I'm not sure that the EMS numbers can reflect patient choices without considering practice size and LSOA population

We agree that list sizes and LSOA population sizes may influence the potential EMS of an LSOA, however we find no general trend in the LSOA-level population size or list size across London in relation to EMS.

13. Page 11: conclusion seems less derived from data and more a commentary on seemingly unaddressed aspects of the online providers, the current PCNs, and the size of CCGs. Can this be written in a way that connects the commentary to the results more directly? Or provide a clearer rationale for why these topics should be addressed in the conclusion?

Thank you. We have amended the conclusion to reflect this.

14. Page 14: figure legends should be more descriptive. The figures are not informative to readers without more context.

Changes have been made to describe figures in more detail.

15. Fig 2: what are the color scales supposed to represent? The scales appear to have different upper bounds in the two images, but the figure implies the scale represents EMS.

The color scale is truncated to an upper limit of 9 to clarify the rest of the distribution owing to its skew

Minor comments

Page 2 lines 32 and 36: the acronym LSOA is never introduced

Thank you – we have now addressed this

Page 4 line 5: should the acronym "NHS" be introduced here formally?

Page 4 line 8: should the acronym "GP" be introduced formally here?

Page 6 line 10: "presentations" is unclear.

Possible typo in the legend of Figure 2: "LSOA I" should probably read "LSOA i"

Thank you – all of the above have been addressed

Reviewer: 3

Reviewer Name: Marieke Perry

Institution and Country: Radboudumc, Nijmegen, The Netherlands

Please state any competing interests or state 'None declared': None declared

Please leave your comments for the authors below

This study aims to identify natural communities of GPs based on patient registration patterns as reaction on the NHS Long Term Plan of Primary Care Network formation. I consider the study design using big data an interesting and original approach to achieving this aim. Since the quality of interprofessional relations is essential in network collaborations, I wonder whether quantitative study alone can service PCN formation.

My statistical knowledge is insufficient to judge the appropriateness of the analyses conducted. I therefore advise an additional statistical review for the study.

Please find below some more detailed comments:

In general, abbreviations are not always appropriately introduced and used throughout the paper.

Changes are made throughout the paper to reflect this.

Abstract:

As a reviewer with limited knowledge of the British healthcare system, the abstract was initially difficult to understand. After reading the full article, everything fell more into place. The abstract would gain clarity when a little more information could be provided on the PCNs. Furthermore, a short explanation of what relevant data these hospital statistics contain. It is mentioned between the lines in the 'main outcome measures section' but it could be made more explicit.

Thank you – we have made some changes to the abstract to clarify this, within the limits of the word count

Introduction:

Well written, clear explanation of NHS policy. Mainly written from a policy perspective. As a clinician, I wonder whether GP could give you the information on the catchment areas of their interprofessional collaboration if you would simply ask them.

This would be one approach, but would only tell us what has been formed, which won't necessarily reflect the underlying registration patterns of patients. As a follow-up to this work, once the formal lists of PCNs have been released, a comparison can be made.

Methods:

Original big data approach, clearly written, but my statistical knowledge is insufficient to judge the appropriateness of the analyses. You describe that outliers are excluded with regard to distance to other practices in a PCN and with regard to PCN size. From a statistical point of view, I understand the choice for exclusion. From a policy point of view, exclusion of practices does not seem a

possibility. The outliers in my view do request additional (qualitative) research, aimed at understanding the mechanisms to the development of these clusters. These considerations could be included in the discussion section.

Thank you. We have clarified this in the limitations section to reflect the fact that there is still a need for practices and policy makers to decide where those with no natural fit should go.

The fact that the study is performed in hospital registrations is in my opinion a major limitation which is considered in the discussion. If the readership that you want to reach with this paper are policy makers and may be also primary care professionals I consider the Equation shown of no additional value. I would consider including these in an Appendix.

Other reviewers have requested the inclusion of additional equations, or have considered them important to interpretation of the manuscript. The decision regarding equation inclusion in the article / appendices will be made according to usual practice of the journal based on their readership.

Results:

The median percentages presented include broad ranges which I would like to see considered in the discussion section.

Thank you, we have added a sentence in to discuss this.

Discussion:

In the summary of the results, you mention that '97.7% of the population can be assigned to a PCN of appropriate size and which are spatially consistent'. To my understanding, this is after the outliers were excluded. How would these results change when these excluded practices were allocated to PCNs as would happen in real life?

This is before exclusion of outliers

The limitation raised on the data source (hospital registrations) I consider a serious limitation. The fact that these data represent more vulnerable patients, because they visited the hospital at least once during the last year, importantly influence the outcomes and would probably only increase the ranges to the median percentages reported. The described mechanism through which the outcomes could be influenced, could be linked to the paragraph on digital NHS general practices.

We agree that this is a limitation and have discussed this in the limitations section. However, primary care level data covering any entire region, of this sort are not available for research purposes within the UK, and this was the best available data source. CPRD is a primary care databased used widely in primary care research, but has patchy geographic coverage which would not be suitable for this analysis. We have edited the text to more explicitly state what we mean by GP practice data.

I would exclude the remark that use of GP data 'was outside the scope of the current study' as you consider the use of GP data preferable.

We feel that this is an important statement to make, as if accessible, this data would be preferable, but unfortunately, is not available for research purposes across all of London. As above we have adjusted this statement in the manuscript.

VERSION 2 – REVIEW

REVIEWER	Christoph Strumann Institute of Family Medicine Universitätsklinikum Schleswig-Holstein, Campus Lübeck Germany
REVIEW RETURNED	29-Apr-2020

GENERAL COMMENTS	The authors have addressed all my previous concerns.
--

REVIEWER	Erika Moen Geisel School of Medicine at Dartmouth
REVIEW RETURNED	22-Apr-2020

GENERAL COMMENTS	The reviewer completed the checklist but made no further comments.
--